# COVID-19 Outcomes in Kidney Transplant Recipients in a German Transplant Center

**DOI:** 10.3390/jcm12186103

**Published:** 2023-09-21

**Authors:** Michael Mikhailov, Klemens Budde, Fabian Halleck, Georgios Eleftheriadis, Marcel G. Naik, Eva Schrezenmeier, Friederike Bachmann, Mira Choi, Wiebke Duettmann, Ellen von Hoerschelmann, Nadine Koch, Lutz Liefeldt, Christian Lücht, Henriette Straub-Hohenbleicher, Johannes Waiser, Ulrike Weber, Bianca Zukunft, Bilgin Osmanodja

**Affiliations:** 1Department of Nephrology and Medical Intensive Care, Charité–Universitätsmedizin Berlin, Charitéplatz 1, 10117 Berlin, Germany; klemens.budde@charite.de (K.B.); fabian.halleck@charite.de (F.H.); georgios.eleftheriadis@charite.de (G.E.); marcel.naik@charite.de (M.G.N.); eva-vanessa.schrezenmeier@charite.de (E.S.); friederike.bachmann@charite.de (F.B.); mira.choi@charite.de (M.C.); wiebke.duettmann@charite.de (W.D.); ellen.von-hoerschelmann@charite.de (E.v.H.); nadine.koch@charite.de (N.K.); lutz.liefeldt@charite.de (L.L.); christian.luecht@charite.de (C.L.); henriette.straub@charite.de (H.S.-H.); johannes.waiser@charite.de (J.W.); ulrike.weber@charite.de (U.W.); bianca.zukunft@charite.de (B.Z.); bilgin.osmanodja@charite.de (B.O.); 2Clinic for Anaesthesiology and Intensive Care Medicine, Charité–Universitätsmedizin Berlin, Campus Benjamin Franklin, Hindenburgdamm 30, 12200 Berlin, Germany

**Keywords:** kidney transplant recipients, immunosuppression, vaccination, SARS-CoV-2, COVID-19, Omicron

## Abstract

Kidney transplant recipients (KTRs) show higher morbidity and mortality from COVID-19 than the general population and have an impaired response to vaccination. We analyzed COVID-19 incidence and clinical outcomes in a single-center cohort of approximately 2500 KTRs. Between 1 February 2020 and 1 July 2022, 578 KTRs were infected with SARS-CoV-2, with 25 (4%) recurrent infections. In total, 208 KTRs (36%) were hospitalized, and 39 (7%) died. Among vaccinated patients, infection with the Omicron variant had a mortality of 2%. Unvaccinated patients infected with the Omicron variant showed mortality (9% vs. 11%) and morbidity (hospitalization 52% vs. 54%, ICU admission 12% vs. 18%) comparable to the pre-Omicron era. Multivariable analysis revealed that being unvaccinated (OR = 2.15, 95% CI [1.38, 3.35]), infection in the pre-Omicron era (OR = 3.06, 95% CI [1.92, 4.87]), and higher patient age (OR = 1.04, 95% CI [1.03, 1.06]) are independent risk factors for COVID-19 hospitalization, whereas a steroid-free immunosuppressive regimen was found to reduce the risk of COVID-19 hospitalization (OR = 0.51, 95% CI [0.33, 0.79]). This suggests that both virological changes in the Omicron variant and vaccination reduce the risk for morbidity and mortality from COVID-19 in KTRs. Our data extend the knowledge from the general population to KTRs and provide important insights into outcomes during the Omicron era.

## 1. Introduction

The World Health Organization (WHO) proclaimed the Coronavirus disease 2019 (COVID-19) as a global pandemic in March 2020 [1]. The rapid emergence and subsequent dominance of new variants on a global scale have become characteristic features of SARS-CoV-2 [2]. The Omicron variant and the time period of its global dominance over other lineages, which can be defined as the Omicron era, relative to previous variants, was characterized by a less severe disease course, lower risk of hospitalization, ICU admission and death [3,4]. In the general population, this is due to a high rate of immunization by means of vaccination and repeated infection, less severe infections due to changes in the virus genome of the Omicron variant, the latter of which are, in part, counterbalanced by a reduction in vaccine effectiveness due to immune escape by the Omicron variants [5]. More than 30 mutations in the spike gene (S) of the Omicron variant were detected in comparison to typically less than 15 in the Alpha and Delta variants [6]. It was assumed that Omicron’s enhanced infectivity was due to increased replication in the bronchi, while its reduced severity was due to reduced penetration into deep lung tissue [7].

Due to their need for systemic immunosuppression, which may impair both cellular and humoral immune responses [8], kidney transplant recipients (KTRs), as well as other organ transplant recipients, show particularly high vulnerability in terms of COVID-19 [9]. KTRs are at high risk of developing a severe disease course with high rates of hospitalization, intensive care unit (ICU) admission, acute kidney injury (AKI), and COVID-19-related mortality [10,11]. COVID-19-related mortality of 14–23% was reported in KTRs [12,13,14,15,16], which was higher than that of the general population, where approximately 1% of patients infected with COVID-19 died until 20 October 2022 [17]. AKI in COVID-19 patients occurs frequently, and two mechanisms are discussed: direct tubular damage by the virus and systemic inflammation and ICU treatment [18,19].

Over time, morbidity and mortality from COVID-19 declined in the general population as well as in KTRs [20,21,22,23]. In KTRs, previous studies showed partially conflicting results regarding the outcomes of COVID-19 during the Omicron era: on the one side, decreased mortality was demonstrated uniformly; on the other side, variable rates of morbidity and hospitalization were reported [23,24,25,26,27]. Since Omicron variants showed increased immune escape, the protective effect of vaccination in KTRs during the Omicron era could be reduced.

Therefore, the objectives of the present study were to report data on COVID-19 disease course and outcomes, as well as to analyze the protective effect of the vaccination against SARS-CoV-2 in KTRs, depending on the predominant variant of concern (VoC).

## 2. Materials and Methods

### 2.1. Data Collection and Extraction

As a main source and interface for data collection, we used our proprietary electronic health record (EHR), TBase, which is fully integrated into the data infrastructure of Charité and enables convenient data acquisition from various data sources [28]. For the investigation of missing data, the EHR of Charité–Universitätsmedizin Berlin (SAP Deutschland SE & Co. KG, Walldorf, Germany) and additional internal databases and medical records were used.

Records of KTRs infected with SARS-CoV-2, as well as further data were identified and extracted using Microsoft Access 2016 (Version 16.0), applying queries with specific criteria, including positive SARS-CoV-2 PCR, positive Anti-SARS-CoV-2-(N)-Ag antibodies, ICD diagnosis U07.1 or U07.2, and medical notes indicating SARS-CoV-2 infection (e.g., positive SARS-CoV-2 rapid test performed by the patient or at a medical facility). The following data were extracted: patient age, sex, date of the last kidney transplantation, dates of infection and reinfection with SARS-CoV-2, dates of COVID-19-related hospital admission and discharge, ICU admission, incidence of acute kidney injury (AKI) as defined by KDIGO, date of death, number of vaccinations against SARS-CoV-2, and immunosuppressive medication at the time of infection with SARS-CoV-2.

Since no routine genetic sequencing was performed and, therefore, exact determination of the causative VoC was not feasible, we chose an indirect definition of VoC. In Germany, a clear predominance of Omicron cases occurred on 10 January 2022 [29]. Consequently, infection before that date was defined as the pre-Omicron era and afterward defined as the Omicron era. Over the course of the pandemic, the following vaccines were administered: BNT162b2 (Comirnaty, BioNTech/Pfizer, Mainz, Germany), mRNA-1273 (Spikevax, Moderna Biotech, Madrid, Spain), ChAdOx1-S (AZD1222, AstraZeneca, Södertälje, Sweden), or Ad26.COV2.S (Johnson & Johnson, Janssen, Beerse, Belgium) [30]. These vaccines were administered based on availability and not using a systematic approach. Therefore, various combinations of vaccine regiments were ultimately administered. Patients who received at least one vaccine dose before infection were defined as being vaccinated.

### 2.2. Analytical Approach

Our primary endpoint was COVID-19-related hospitalization. Secondary endpoints were COVID-19-related death, AKI, and ICU admission.

The causal directed acyclic graphs (DAGs) in Figure A1 summarize our assumptions regarding the causal relationship of important measured and unmeasured variables, which are discussed in detail in Section A.1.

No imputation methods were used, and a complete case analysis was performed.

### 2.3. Statistical Analysis

All of the statistical analyses were performed using IBM SPSS Statistics (Version 28.0.1.0). First, we described the frequency of each outcome depending on the era of infection and vaccination status. Continuous variables were tested for normal distribution using a Shapiro–Wilk test and compared using binomial logistic regression analysis, the Mann–Whitney U test and Spearman’s correlation. Categorical variables were compared using the Chi-squared-test. Based on the assumptions above and the measured variables available, we included age, immunosuppression at the time of infection, time since transplantation, and era of infection as potential confounders when estimating the effect of vaccination on COVID-19 outcomes (hospitalization, COVID-19-related mortality, AKI, and ICU admission) by performing multinomial logistic regression. A *p*-value of 0.05 or less (two-tailed) was considered statistically significant.

The ethics committee of Charité–Universitätsmedizin Berlin approved this study (EA1/030/22).

## 3. Results

From approximately 2500 KTRs followed-up at our institution [31], 578 (23%) were identified to be infected with SARS-CoV-2 between 1 February 2020 and 1 July 2022. In total, 208 KTRs (36%) were hospitalized, and 39 (7%) died from COVID-19. Additionally, 10 patients died from other causes than COVID-19: infection (three patients), malignancy (two patients), sudden death (two patients), and hemorrhage (three patients). In total, 338 (58%) KTRs were vaccinated against SARS-CoV-2 prior to the infection, with a median number of three vaccination doses. The hospitalization rate for COVID-19 started to decrease at the end of the Delta wave in November and December 2021 and further decreased after the Omicron variant became dominant (Figure 1 and Figure 2). Infection, hospitalization, and mortality trends, as well as demographics, course of disease and outcomes of COVID-19 are detailed in Figure 1, Figure 2 and Table 1, respectively.

At our center, 317 (55%) and 261 (45%) infections occurred in the pre-Omicron and Omicron era, respectively. Hospitalization rate decreased from 49% to 21% from the pre-Omicron to the Omicron era (*p* < 0.001), and mortality decreased from 10% to 3% (*p* < 0.001). Likewise, the proportion of patients admitted to the ICU decreased from 18% to 7% (*p* < 0.001), and the number of patients with AKI decreased from 21% to 13% (*p* = 0.022). The differences in disease course and outcomes between the pre-Omicron and Omicron eras are summarized in Table 2.

Overall, 240 patients were not vaccinated before the infection with SARS-CoV-2, of which 205 (85%) were infected during the pre-Omicron and 35 (15%) during the Omicron era. Patients who were vaccinated before infection showed reduced reinfection rates (2% vs. 8%, *p* = 0.002), reduced hospitalization rates (24% vs. 53%, *p* < 0.001), reduced mortality rates (4% vs. 10%, *p* = 0.003), were less likely to be admitted to the ICU (9% vs. 17% *p* = 0.008), and were less likely to develop AKI (11% vs. 28%, *p* < 0.001) in comparison to unvaccinated patients.

Table 3 summarizes the data on COVID-19 disease course and outcomes depending on vaccination status. The protective effects of vaccination were more pronounced for the Omicron era than for the pre-Omicron era, which is summarized in Table 4. Unvaccinated patients infected in the Omicron era show mortality (9% vs. 11%) and morbidity (hospitalization 52% vs. 54%, ICU admission 12% vs. 18%) comparable to the pre-Omicron era, while vaccinated patients had improved outcomes, especially in the Omicron era with respect to mortality (2%) and morbidity (hospitalization, 16%; ICU admission, 6%). A logistic regression demonstrated that the number of vaccination doses was inversely correlated with mortality (OR 0.71, 95% CI [0.56–0.89]), hospitalization (OR 0.65, 95% CI [0.58–0.73]), ICU admission (OR 0.76, 95% CI [0.65–0.90]) and AKI (OR 0.73, 95% CI [0.63–0.84]). Figure 3 demonstrates the changes in hospitalization incidence with the increasing number of vaccination doses (applied before the first infection with SARS-CoV-2).

To estimate the effect of vaccination on COVID-19-related outcomes, we adjusted for potential confounding due to era of infection, age, immunosuppression, and time since transplantation (cf. Figure A1 and Item A1) in the multivariable regression. We found that being unvaccinated (OR = 2.15, 95% CI [1.38, 3.35]), infection in the pre-Omicron era (OR = 3.06, 95% CI [1.92, 4.87]) and higher patient age (OR = 1.04, 95% CI [1.03, 1.06]) were independent risk factors for COVID-19 hospitalization. For COVID-19-related death, only infection in the pre-Omicron era (OR = 3.08, 95% CI [1.14, 8.33]) and higher patient age (OR = 1.10, 95% CI [1.06, 1.14]) were independent risk factors. Being unvaccinated (OR = 2.72, 95% CI [1.55, 4.77]) and higher patient age (OR = 1.03, 95% CI [1.02, 1.05]) were independent risk factors for COVID-19-related acute kidney injury. For ICU admission, infection in the pre-Omicron era (OR = 3.22, 95% CI [1.63, 6.37]) and higher patient age (OR = 1.06, 95% CI [1.03, 1.08]) were independent risk factors. A steroid-free immunosuppressive regimen was found to be a protective factor both against hospitalization for COVID-19 (OR = 0.51, 95% CI [0.33, 0.79]), COVID-19-related acute kidney injury (OR = 0.45, 95% CI [0.26, 0.80]) and ICU admission (OR = 0.43, 95% CI [0.22, 0.83]) (Figure 4). Other immunosuppressive agents (mycophenolic acid, tacrolimus, cyclosporine, belatacept, mTOR inhibitors, and azathioprine) were not significantly associated with COVID-19 mortality or morbidity.

We furthermore analyzed the rejection rates in both eras. In the pre-Omicron era, 88 rejection episodes occurred over the 1.94-year observation period, while in the Omicron era, 28 rejection episodes occurred over the 0.47-year observation period. Based on the population size of 2500 patients, the rejection rates were 1.81/100 patient-years in the pre-Omicron era and 2.38/100 patient-years in the Omicron era. At the same time, a small increase in biopsies occurred (5.5 biopsies/100 patient-years in the pre-Omicron era vs. 7.2 biopsies/100 patient-years in the Omicron era).

## 4. Discussion

In the present article, we show that hospitalization rate and COVID-19-related mortality and morbidity in KTRs with COVID-19 decreased since the beginning of the pandemic, but especially after the Omicron variant became dominant, and a higher proportion of patients were vaccinated. We further show that both infection in the pre-Omicron era and being unvaccinated are independent risk factors for COVID-19-related hospitalization in KTRs. Other factors affecting disease severity, assessed in terms of hospitalization and COVID-19-related death, included patient age and steroid intake. Other immunosuppressive agents did not affect disease severity in our analysis.

Additionally, the number of vaccinations was inversely correlated with morbidity and mortality, further underlining the protective effect of vaccination in KTRs.

These findings are in line with observations in the general population, where both vaccination and Omicron variant reduced the risk of severe COVID-19, independently [5]. Within our cohort, we could also confirm that patient age and steroid intake pose risk factors for severe COVID-19, which has been described before in the general population as well as in other cohorts of KTRs [24,32,33,34].

Our findings highlight that mortality and morbidity in unvaccinated KTRs remain unchanged after Omicron dominance, underlining the ongoing need for vaccination or alternative immunization strategies in this vulnerable group of patients [35]. Vaccination significantly reduced the rate of hospitalization and disease-specific morbidity and mortality during the Omicron era.

The reinfection rate in patients with COVID-19 has not been studied before, probably due to the rather low frequency of reinfections and small cohorts. We found the reinfection rate to be strongly associated with vaccination status, with the majority of reinfections occurring in unvaccinated patients, which is in line with the existing literature for the general population [36,37]. The overall infection rate in KTRs was smaller (23%) than for the general population, where approximately 32% were infected until 17 June 2022, according to [38]. Since the estimated number of undetected cases is probably higher in the general population, where testing is less likely to be performed in mild cases, we hypothesize that KTRs are more cautious and avoid potential infections more successfully than the general population.

Rejection rates were slightly increased in the Omicron era in comparison to the pre-Omicron era, which is mostly explained by an increase in biopsies performed. This suggests that virological changes did not influence rejection rates in KTRs but that admission policies due to institutional restrictions and patient preferences were more important.

### Strengths and Limitations

This is one of the first studies to analyze disease outcomes in KTRs during the Omicron era. The incidence of COVID-19 and clinical outcomes after infection were assessed in a large cohort of closely monitored KTRs with granular data on outcome and vaccination status. Since most KTRs with COVID-19 were followed-up at our institution using a telemedicine approach and were hospitalized at our center if necessary, the ground truth of the underlying data on hospitalization, ICU admission, AKI, and mortality is high [39].

The major limitation is that therapeutic measures, such as treatment with monoclonal antibodies or antiviral therapies, like remdesivir, have not been included into our analysis [40,41]. Consequently, the decrease in disease severity is not solely explained by vaccination and virological changes but in part due to the increased availability and use of therapeutic agents. Another limitation concerns the power of this study. Especially with respect to mortality, the event rate was rather low (*n* = 39), which reduces the reliability of the results from multivariable analysis for this endpoint. Since we only analyzed the influence of single immunosuppressive agents on COVID-19 disease severity, we could not exclude that certain combinations of immunosuppressive agents may influence the risk. However, since most patients received a calcineurin inhibitor and mycophenolic acid with or without steroids as maintenance immunosuppression, the reliability of analyses for the other immunosuppressive agents is reduced due to the low patient count.

## 5. Conclusions

Disease severity and mortality from COVID-19 are substantially reduced in vaccinated KTRs infected during the Omicron era. Unvaccinated KTRs show high mortality and morbidity independent of the causative variant.

## Figures and Tables

**Figure 1 jcm-12-06103-f001:**
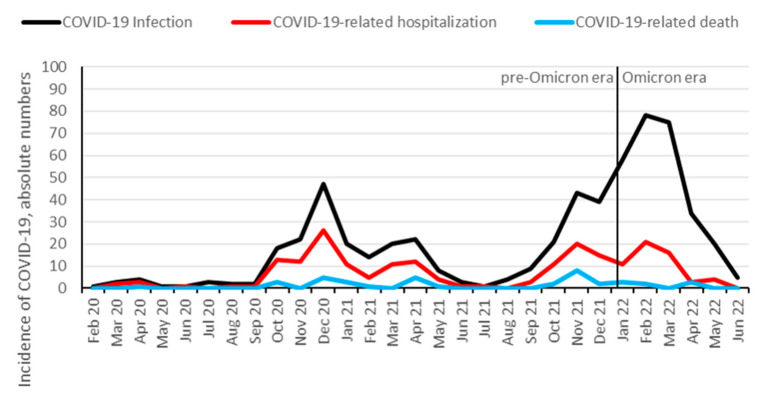
Absolute numbers of COVID-19 infection, hospitalization, and mortality over time among KTR. Data are aggregated per month from February 2020 until June 2022. Omicron dominance occurred on 10 January 2022.

**Figure 2 jcm-12-06103-f002:**
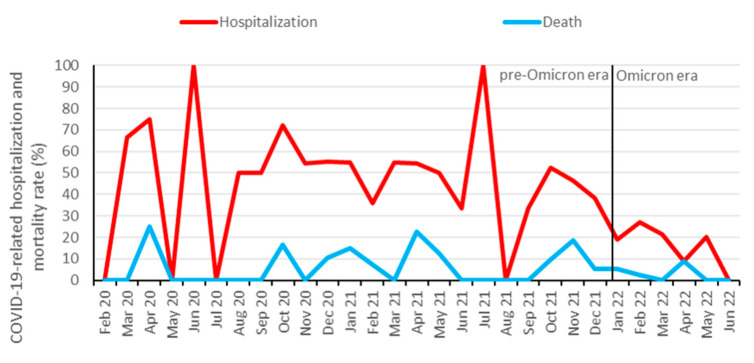
Hospitalization and mortality rate of COVID-19 among KTRs. Data are aggregated per month from February 2020 to June 2022. Omicron dominance occurred on 10 January 2022. During low incidence periods from May to August 2020 and June to August 2021, higher fluctuations in hospitalization rates were observed.

**Figure 3 jcm-12-06103-f003:**
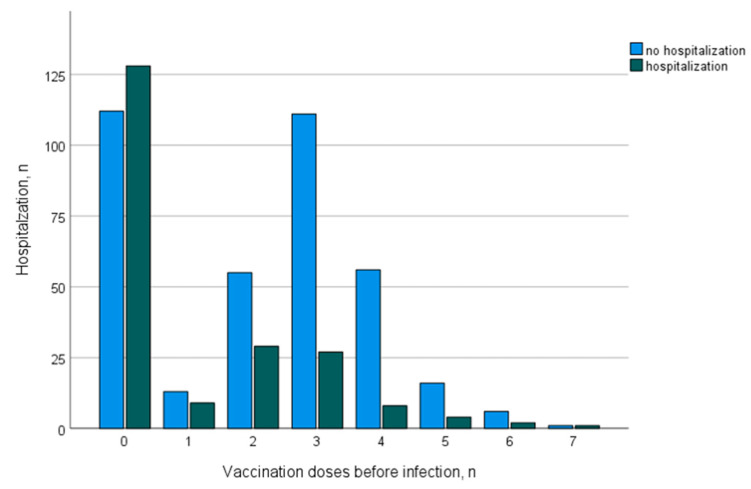
Distribution of hospitalization incidence depending on the number of vaccination doses.

**Figure 4 jcm-12-06103-f004:**
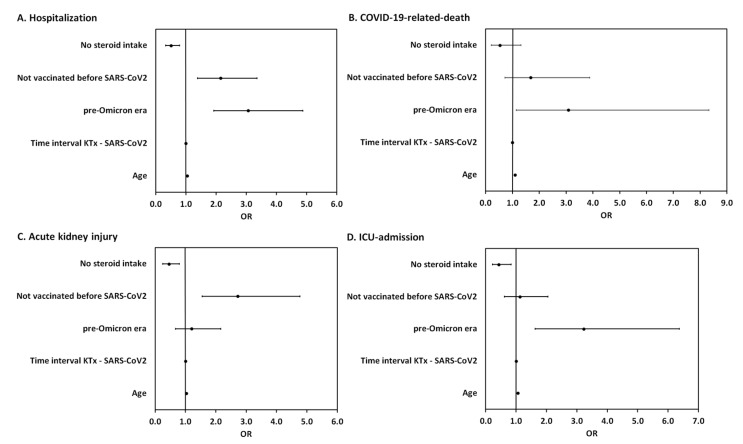
Forest plot summarizing the odds ratios of steroid intake, vaccination status, era of infection, time since transplantation, and patient age for hospitalization (**A**), COVID-19-related death (**B**), acute kidney injury (**C**) and ICU admission (**D**) in KTRs with COVID-19 from February 2020 to June 2022.

**Table 1 jcm-12-06103-t001:** Demographics of KTRs, COVID-19 disease course and outcomes.

KTRs infected with COVID-19, *n*	578
Reinfection, *n* (%)	25 (4)
Males, *n* (%)	355 (61)
Median age in years (IQR)	54.17 (19.24–85.57)
Median time post-KTx to COVID-19 in months (IQR)	101 (0–1463)
Vaccination before the infection, *n* (%)	338 (58)

Immunosuppressive therapy at COVID-19 diagnosis	
Mycophenolic acid, *n* (%)	481 (83)
Tacrolimus, *n* (%)	379 (66)
Cyclosporine, *n* (%)	110 (19)
Steroids, *n* (%)	373 (65)
Belatacept, *n* (%)	27 (5)
mTOR inhibitors, *n* (%)	14 (2)
Azathioprine, *n* (%)	18 (3)
Patients with missing data about IS, *n* (%)	35 (6)

COVID-19 clinical course and management	
Acute kidney injury	103 (18)
Hospitalization, *n* (%)	208 (36)
Median duration in days (IQR)	11 (1–123)
ICU admission, *n* (%)	73 (13)
COVID-19 outcomes	
Death, *n* (%) ^1^	39 (7)

^1^ COVID-19-related death; IQR—interquartile range; KTx—kidney transplantation; KTR—kidney transplant recipients; IS—immunosuppression.

**Table 2 jcm-12-06103-t002:** Era-specific COVID-19 disease course and outcomes.

	Pre-Omicron ^1^	Omicron ^2^	*p*
KTRs infected with COVID-19, *n* (%)	317 (55)	261 (45)	
Reinfection, *n* (%)	23 (7)	2 (1)	
Vaccination before the infection, *n* (%)	110 (35)	228 (87)	

COVID-19 clinical course and management			
Acute kidney injury	68 (21)	35 (13)	0.022
Hospitalization, *n* (%)	154 (49)	54 (21)	<0.001
Median duration in days (IQR)	12 (1–123)	11 (1–103)	<0.001
ICU admission, *n* (%)	56 (18)	17 (7)	<0.001
Death, *n* (%) ^3^	31 (10)	8 (3)	0.001

^1^ before 10 January 2022; ^2^ from 10 January 2022; ^3^ COVID-19-related death; KTR—kidney transplant recipients; IQR—interquartile range.

**Table 3 jcm-12-06103-t003:** COVID-19 disease course and outcomes of not vaccinated^1^ and vaccinated ^1^ KTRs ^2^.

	Not Vaccinated	Vaccinated	*p*
Total, *n*	240	338	
Reinfection, *n* (%)	18 (8)	7 (2)	0.002
Omicron ^3^, *n*(%)	35 (15)	226 (67)	<0.001

COVID-19 clinical course and management			
Acute kidney injury	66 (28)	37 (11)	<0.001
Hospitalization, *n* (%)	128 (53)	80 (24)	<0.001
Median duration in days (IQR)	11 (1–123)	12 (1–123)	<0.001
ICU admission, *n* (%)	41 (17)	32 (9)	0.008
Death, *n* (%) ^4^	25 (10)	14 (4)	0.003

^1^ prior to the SARS-CoV-2 infection; ^2^ kidney transplant recipients ^3^ from 10 January 2022; ^4^ COVID-19-related death. KTRs—kidney transplant recipients; ICU—intensive care unit; IQR—interquartile range.

**Table 4 jcm-12-06103-t004:** Era- and vaccination-specific COVID-19 disease outcomes.

	Pre-Omicron ^1^		Omicron ^2^	
	Vaccinated	Not Vaccinated	Vaccinated	Not Vaccinated
KTRs infected with COVID-19, *n* (%) ^3^	110 (35)	207 (65)	228 (87)	33 (13)
Reinfection, *n* (%)	5 (5)	18 (9)	2 (1)	0 (0)

COVID-19 outcomes				
Acute kidney injury	16 (15)	52 (25)	21 (9)	14 (42)
Hospitalization, *n* (%)	43 (39)	111 (54)	37 (16)	17 (52)
Duration in days (IQR)	14 (1–123)	11 (1–123)	11 (1–86)	12 (3–103)
ICU admission, *n* (%)	19 (17)	37 (18)	13 (6)	4 (12)
Death, *n* (%) ^4^	9 (8)	22 (11)	5 (2)	3 (9)

^1^ before 10 January 2022; ^2^ from 10 January 2022; ^3^ of all pre-Omicron/Omicron patients; ^4^ COVID-19-related death. KTRs—kidney transplant recipients; IQR—interquartile range.

## Data Availability

Not applicable.

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
