# Peer review of "COVID-19 Outcomes in Kidney Transplant Recipients in a German Transplant Center"

_jcm, 2023, doi:10.3390/jcm12186103_

Round 1
Reviewer 1 Report
The present study aims to describe clinical outcomes in KTRs affected by Covid-19 infection. Overall, the manuscript is well written, although the novelty of the research is limited and other studies addressed this topic. One of the main limitation (already reported by authors is the absence of information on antiviral and/or monoclonal antibodies administered in the Omicron era compared to the first pandemic wave). Few additional comments:
- I would improve introduction adding some data on AKI in KTRs in Covid and potential molecular mechanisms involved (see for example PMID: 33656736 and PMID: 37239144).
- Methods should be expanded
- Figure 1 is poorly readable; I could suggest to put it in the supplementary files
- What is the impact of immunosuppressive therapies? In other words, are outcomes similar between different immunosuppressive regimens (for example a regimen with 2 drugs vs a regimen with 3 drugs)? what about m-TOR inhibitors?
- the discussion should be expanded too, with more comments on the impact of immunosuppressive therapies in clinical outcomes in this population
Few minor editing required
Author Response
Dear Reviewer,
thank you very much for your valuable comments! Here we provide point-by-point response to them:
- I would improve introduction adding some data on AKI in KTRs in Covid and potential molecular mechanisms involved (see for example PMID: 33656736 and PMID: 37239144).
- The AKI-references added to the manuscript, Introduction section.
- Methods should be expanded.
- We expanded the methods section by further defining the criteria, which were applied to identify SARS-CoV-2 positive patients. We added an explanation regarding vaccination regimens.
-
Figure 1 is poorly readable; I could suggest to put it in the supplementary files.
- Figure 1 moved to the Appendix section.
- What is the impact of immunosuppressive therapies? In other words, are outcomes similar between different immunosuppressive regimens (for example a regimen with 2 drugs vs a regimen with 3 drugs)? what about m-TOR inhibitors?
-
We analyzed, whether any immunosuppressive agent (mycophenolic acid, tacrolimus, cyclosporine, belatacept, mTOR inhibitors, azathioprine) were associated with negative outcomes, and only found that steroid treatment was significantly associated with such outcomes. We added a sentence to the manuscript where we mentioned the non-significant results for immunosuppressive regimens.
-
-
The discussion should be expanded too, with more comments on the impact of immunosuppressive therapies in clinical outcomes in this population.
-
We added a summary of the additional results with respect to immunosuppressive agents to the Results section and some limitations regarding these analyses to the Discussion section.
-
Reviewer 2 Report
Dear Authors,
I read with interest the paper " COVID-19 Outcomes in Kidney Transplant Recipients in a German Transplant Center ". In this study a comprehensive overview of COVID-19 course and outcomes in a frail population, as kidney transplantation recipients, from February 2020 to June 2022. Overall the paper is interesting and authors should be commended for their work.
Few remarks:
- Did you check if there is a correlation between the number of years from transplantation and severity of infection?
- In the discussion section you mentioned the telemedicine approach for your consultation. It is a methods currently adopted in your center? COVID-19 shows us as this tool can be as effective as in person consultations, limiting the possibility of infections without delaying the follow-up of our patients. this is an example of telemedicine applied in an Urological Italian High volume center during COVID-19 pandemic: 10.1007/s00345-020-03536-x.
Author Response
Dear Reviewer,
thank you very much for your valuable comments! Here we provide point-by-point response to them:
- Did you check if there is a correlation between the number of years from transplantation and severity of infection?
- There was no significant correlation between number of years from transplantation and COVID-19 severity. We already mentioned this parameter (“time since transplantation”) as a potential confounder for the effect of vaccination on disease severity (section Results, p. 8) and decided not to comment on the individual statistical correlation in the revised manuscript.
- In the discussion section you mentioned the telemedicine approach for your consultation. It is a methods currently adopted in your center? COVID-19 shows us as this tool can be as effective as in person consultations, limiting the possibility of infections without delaying the follow-up of our patients. this is an example of telemedicine applied in an Urological Italian High volume center during COVID-19 pandemic: 10.1007/s00345-020-03536-x.
- Yes, we still a telemedicine approach for all KTR with COVID-19. This helps us to give quick therapy recommendations to the patients without the risk of disease transmission.
Round 2
Reviewer 1 Report
No further comments. The manuscript is suitable for publication according to journal's priority